# Physiological and Genetic Modifications Induced by Plant-Growth-Promoting Rhizobacteria (PGPR) in Tomato Plants under Moderate Water Stress

**DOI:** 10.3390/biology12070901

**Published:** 2023-06-23

**Authors:** Jose Antonio Lucas, Ana Garcia-Villaraco, Maria Belen Montero-Palmero, Blanca Montalban, Beatriz Ramos Solano, Francisco Javier Gutierrez-Mañero

**Affiliations:** Plant Physiology, Pharmaceutical and Health Sciences Department, Faculty of Pharmacy, Universidad San Pablo-CEU, CEU-Universities, 28668 Boadilla del Monte, Spain; anabec.fcex@ceu.es (A.G.-V.); mariabelen.monteropalmero@ceu.es (M.B.M.-P.); bramsol@ceu.es (B.R.S.); jgutierrez.fcex@ceu.es (F.J.G.-M.)

**Keywords:** water stress, PGPR, PEG6000, oxidative stress, photosynthesis efficiency, *P5CS*, *NCDE1*, *plasma membrane ATPase*

## Abstract

**Simple Summary:**

The availability of water is an essential factor in food production and, therefore, for food security, since even with moderate water deficits, plants reduce their production significantly. In a context of climate change, in which water scarcity is becoming increasingly evident, food security can be seriously compromised. In this work, the use of plant-growth-promoting rhizobacteria (PGPR) strains is proposed as a biotechnological tool to improve the physiological conditions of tomato plants under conditions of moderate water stress with the aim of trying not to lose productive capacity under these conditions. To validate it, the effect of PGPR on photosynthesis, oxidative state, and innate defensive response were determined in plants growing under moderate water stress. The results obtained show that PGPR can alter the physiology of plants in different ways, improving their resistance under moderate water stress conditions, maintaining their productive capacity. The use of this technology in agricultural production could reduce water consumption without reducing food production, which would have a strong social impact at all levels.

**Abstract:**

Physiological, metabolic, and genetic changes produced by two plant growth promoting rhizobacteria (PGPR) *Pseudomonas* sp. (internal code of the laboratory: N 5.12 and N 21.24) inoculated in tomato plants subjected to moderate water stress (10% polyethylene glycol-6000; PEG) were studied. Photosynthesis efficiency, photosynthetic pigments, compatible osmolytes, reactive oxygen species (ROS) scavenging enzymes activities, oxidative stress level and expression of genes related to abscisic acid synthesis (ABA; 9-cis-epoxycarotenoid dioxygenase *NCDE1* gene), proline synthesis (Pyrroline-5-carboxylate synthase *P5CS* gene), and plasma membrane ATPase (*PM ATPase* gene) were measured. Photosynthetic efficiency was compromised by PEG, but bacterial-inoculated plants reversed the effects: while N5.12 increased carbon fixation (37.5%) maintaining transpiration, N21.24 increased both (14.2% and 31%), negatively affecting stomatal closure, despite the enhanced expression of NCDE1 and plasma membrane ATPase genes, evidencing the activation of different adaptive mechanisms. Among all parameters evaluated, photosynthetic pigments and antioxidant enzymes guaiacol peroxidase (GPX) and ascorbate peroxidase (APX) responded differently to both strains. N 5.12 increased photosynthetic pigments (70% chlorophyll a, 69% chlorophyll b, and 65% carotenoids), proline (33%), glycine betaine (4.3%), and phenolic compounds (21.5%) to a greater extent, thereby decreasing oxidative stress (12.5% in Malondialdehyde, MDA). Both bacteria have highly beneficial effects on tomato plants subjected to moderate water stress, improving their physiological state. The use of these bacteria in agricultural production systems could reduce the amount of water for agricultural irrigation without having a negative impact on food production.

## 1. Introduction

Water is a crucial resource for agriculture, representing 72% of the total volume of fresh water consumed. Severe water stress is considered when, on average, more than 40% of the available water supply is used each year and where the demand for water is higher than the amount available during a certain period of time. Spain is one of the 44 countries highly affected by water stress [1].

In Spain, the agricultural sector accounts for around 80% of the total water consumption. Irrigation is a fundamental element of our agri-food system. In Spain, it represented 22.9% of the cultivated area in 2021 (7.8% of the total geographical area), contributing to 50% of the final crop production. Therefore, irrigation is of paramount importance to keeping Spain as the leading exporter of fruit and vegetables in the EU, and to be among the three leading exporters in the world, along with China and the United States.

To date, the water scarcity problem has been addressed almost exclusively by improving irrigation systems. However, at no time has it been contemplated to reduce water consumption because, as indicated above, this has negative consequences on crop production.

Drought stress, even moderate, has negative effects on plant growth and productivity. A water deficit reduces cell division and elongation, alters tissue water status, causes stomatal closure, and damages the photosynthetic machinery, limiting photosynthetic activity [2,3]. In addition, drought stress increases the production of reactive oxygen species (ROS), such as singlet oxygen (^1^O_2_), hydroxyl radicals (OH^−^), superoxide anion (O_2_^−^), and hydrogen peroxide (H_2_O_2_), which cause lipid peroxidation of cell membranes, protein denaturation, and nucleic acid destruction [4,5]. Plants have developed tolerance mechanisms that may promote their survival in water-stressed environments [3,6]. The production of ROS induced by abiotic stress activates signalling mechanisms in plants that induce hormonal changes, mainly an increase in endogenous ABA, and activate genes to produce osmolytes and antioxidant enzymes [7,8]. Compatible solutes such as proline, glycine betaine, soluble proteins, and sugars accumulate in plants in response to drought, improving water potential. In addition, the activity of antioxidants, both enzymatic and non-enzymatic, is increased under drought stress to eliminate excess ROS and, thus, protect membranes and organic molecules [2,9]. In this study, a biotechnological approach to maintaining plant productivity under conditions of moderate water stress based on the use of beneficial bacteria is proposed.

Some bacteria that live in the roots of plants are known to have beneficial effects on plants. Beneficial soil bacteria, known as plant-growth-promoting rhizobacteria (PGPR), live in free symbiosis with plants in the rhizosphere where they play a critical role in helping plants overcome abiotic stress and remain productive [10]. This is because PGPR induce a physiological alert state that keeps the plant’s defense mechanisms on alert, a phenomenon known as priming. In this state, the plant responds more quickly and/or more strongly to stress, resulting in better stress tolerance compared to naïve (non-primed) plants [11,12]. Specific PGPR have been shown to be an enormous benefit for improving plant fitness under these stress conditions, to the point that the term induced systemic tolerance (IST) has been coined to define the plant metabolic situation induced by these PGPR [3,13,14].

PGPR are able to bind plant receptors, triggering a systemic response in the aboveground parts of the plant. An ROS burst follows plant–PGPR recognition, therefore altering the upstream signaling, triggering the modification of gene expression that results in an increased metabolic adaptation that best serves the plant. Abiotic stress also generates an ROS burst; PGPR can also modulate abiotic stress responses, improving plant adaptation. Therefore, the ROS burst is common to the perception of any stimuli by plants [15,16]. It becomes evident that plant metabolism can be modulated through the microbiome, boosting plant innate responses to stress, that is, enhancing adaptive metabolism [17].

Other mechanisms normally used by some PGPR to systemically improve the physiological state of plants in situations of water stress is the regulation of hormonal balances, especially that of ethylene and ABA, the two hormones related to triggering resistance to abiotic stress situations [3]. The relationship of some PGPRs with the decrease in ethylene concentration in the plant due to their ability to use the ethylene precursor, ACC, as a nitrogen source, has been well-studied [18]. However, there are few studies on the relationship of PGPR with the ABA balance in the plant [19].

Based on what was indicated above, the objective of the work has been to evaluate the ability of two PGPR to improve resistance to moderate water stress. For this, the physiological and genetic responses of plants subjected to moderate water stress, inoculated with PGPR, were studied at the physiological, metabolic, and genetic levels. The response of photosynthetic activity was studied. In addition, markers related to the resistance to this type of stress were analyzed separately in the root and aerial parts: the accumulation of compatible osmolytes, enzymatic activities related to ROS scavenging, and markers of oxidative stress. At the genetic level, the differential expression of genes that could be related to the response of plants to the stress situation dealing with the ISR, proline synthesis, and ABA response was studied. The two PGPR strains used were chosen because, in a previous work by the group [20], we demonstrated their capacity to improve *Arabidopsis thaliana* resistance to *Pseudomonas syringae* DC3000 infection.

## 2. Material and Methods

### 2.1. Strains Used

Two *Pseudomonas* sp strains were used in this experiment. Strains were isolated from *Nicotiana glauca* rhizospheric soil [21] and have not been subjected to mutation processes or genetically manipulated. Both were able to produce siderophores and solubilize phosphate. In addition, both protected tomato plants against attack of pathogen *Pseudomonas syringae* DC3000, even above plants treated with BTH [20]. These strains were named as N 5.12 and N 21.24; their 16s RNA sequences are deposited in GenBank database under the accession number MH571507 and MH571660, respectively.

### 2.2. Experimental Design

Experimental design is shown in Figure 1. Seeds of *Solanum lycopersicum* var. Casillas were sown in trays (12) of 28 alveoli of 150 cm^3^ each. During all experiment, each alveolus was watered with 20 mL of tap water each for two days and were grown in a culture chamber (Sanyo MLR-350H) with a 12 h light (350 µE/m^2^ s at 27 °C) and 12 h dark period (22 °C) at 70% relative humidity. Forty days after sowing, when plants had two pairs of leaves, 84 seedlings (three trays) were inoculated with N 5.12 and another 84 seedlings (three trays) with N 21.24. Each tray constituted a replicate. Strains were inoculated by soil drench with 5 mL of a suspension of bacterial cells, grown for 24 h in nutrient broth (CONDA) at 28 °C, and adjusted to a density of 10^8^ cfu/mL. One week after the first inoculation, seedlings were inoculated again. One week after the second inoculation, 9 trays were subjected to water stress (6 trays inoculated with bacteria and 3 trays non inoculated). To subject the plants to moderate water stress, they were irrigated with polyethylene glycol-6000 (PEG) at 10% (osmotic potential of −0.54 MPa) and the other 3 were watered with tap water (control). Three days after, photosynthesis efficiency was measured. Roots and aerial parts were harvested separately, powdered in liquid nitrogen, and stored at −80 °C. These plant samples were used for different analysis and for gene expression analysis by qPCR. In total, 4 treatments were established: Control, PEG, PEG + N 5.12, and PEG + N 21.24.

### 2.3. Net Photosynthetic Rate (PN) and Transpiration Rate (E) Measurements

Net photosynthetic rate (PN) and transpiration rate (E) were measured simultaneously by CI-340 portable photosynthesis system (CID Bio-Science, Inc., Camas, WA, USA) in 10 plants of each tray. In addition, the water use efficiency (WUE = Pn/E) was calculated. CI-340 system was equipped with the light control module CI-301 LA to adjust the light intensity in 500 μmol/m2 s. Net photosynthetic rate is determined by measuring CO_2_ before and after it enters the leaf chamber to calculate the rate of CO_2_ assimilation by a known leaf area (μmol/m2 s). Transpiration rate is determined by measuring water vapor before and after it enters the leaf chamber to calculate the rate of water vapor flux per one-sided leaf area (mmol/m^2^ s).

### 2.4. Chlorophyll Fluorescence Measurements

Chlorophyll fluorescence was measured with a pulse-amplitude-modulated (PAM) fluorometer (HansatechFM2, Hansatech, Inc., Norfolk, UK) on 1 h dark-adapted leaves. Minimal fluorescence (Fo; dark-adapted minimum fluorescence) was measured with weak modulated irradiation (1 mol/m^2^ s). Maximum fluorescence (Fm) was determined for the dark-adapted state by applying a 700 ms saturating flash (9000 mol/m^2^ s). The variable fluorescence (Fv) was calculated as the difference between the maximum fluorescence (Fm) and the minimum fluorescence (Fo). The maximum photosynthetic efficiency of photosystem II (maximal PSII quantum yield) was calculated as Fv/Fm. Immediately, the leaf was continuously irradiated with red–blue actinic beams (80 mol/m^2^ s) and equilibrated for 15 s to record Fs (steady-state fluorescence signal). Then, with plants adapted to light conditions, Fm’ (maximum fluorescence) was measured by applying another saturation flash (9000 mol/m^2^ s). Finally, φPSII (PSII quantum yield φPSII = (Fm’ − Fs)/Fm’) and NPQ (non-photochemical quenching coefficient NPQ = (Fm − Fm’)/Fm’) were determined.

### 2.5. Photosynthetic Pigments (Chlorophylls and Carotenoids) Determination

To determine photosynthetic pigments, 0.1 g of powder was mixed vigorously with 1 mL of 80% acetone. After 16 h at 4 °C, tubes were vortexed and centrifugated at 4000 rpm for 2 min to clarify the supernatant. One hundred microliters of the supernatant were mixed with 900 μL of 80% acetone. Absorbances at 663, 647, and 470 nm were measured and with these data were calculated the concentration of Chlorophyll a (Chl a), chlorophyll b (Chl b), and carotenoids following the equations proposed by Porra et al. [22].
Chl a = [ (12.25 × Abs663) − (2.79 × Abs647)] × V/weight (mg)
Chl b = [(21.50 × Abs647) − (5.10 × Abs663)] × V/weight (mg)
Carotenoids = [(1000 × Abs470) − (1.82 × Chl a) − (85.02 × Chl b)/198] × V/weight (mg)

### 2.6. Glycine Betaine Determination

Glycine betaine determination was carried out following the method of Veladez-Bustos et al. [23]. One hundred mg of powder was mixed with 1.5 mL of 2N H_2_SO_4_ and the mixture was heated up to 60 °C for 10 min. Afterwards, the mixture was centrifuged at 14,000 rpm for 10 min at room temperature. One hundred and twenty-five μL of the supernatant was mixed in a new Eppendorf with 50 μL cold KI-I_2_ (15.7 g of iodine and 20 g of KI in 100 mL of sterilized water). KI-I_2_ causes the precipitation of glycine betaine in the form of golden crystals. After adding KI-I_2_, the next steps were to be performed under darkness. The tubes were stored at 4 °C for 16 h and then centrifuged at 14,000 rpm for 30 min at 0 °C. Tubes must be maintained in ice. After carefully eliminating the supernatant, the precipitate was mixed with 1.4 mL of 1,2-dichloroethane, and, finally, the absorbance was read using a spectrophotometer at 290 nm for 48 h after adding 1,2-dicholoroethane. Concentration was expressed as mg/g.

### 2.7. Proline Content Determination

Proline content was determined according to the method of Wang et al. [24] with some modifications. Samples (0.2 g) were mixed with 1 mL of 3% (*w*/*v*) sulphosalicylic acid and centrifuged at 1000× *g* for 5 min at 4 °C. One hundred mL of supernatant was mixed with 0.1 mL of 3% (*w*/*v*) sulphosalycilic acid, 0.2 mL of 96% (*v*/*v*) acetic acid, and 0.2 mL of 1% (*w*/*v*) ninhydrin, and incubated in a water bath at 96 °C. After 1 h, the mixtures were vortexed for 1 min with 1 mL toluene and centrifuged at 1000× *g* for 5 min at 4 °C. Absorbance of supernatant was measured at 520 nm and proline content was calculated using an extinction coefficient of 0.9986 mM^−1^/cm. Concentration was expressed as mM/g.

### 2.8. Soluble Sugars Determination

Soluble sugars were quantified following the procedure of Yemm and Willis [25]. One hundred mL of the plant ethanolic extract were mixed with 3 mL of anthrone reagent freshly prepared (200 mg of anthrone with 100 mL of 72% sulfuric acid). After an incubation at 100 °C for 10 min, samples were cooled in ice, and 250 μL of each were put in the 96 well-plate, and absorbance was measured at 620 nm (SPECTROstar nano). To calculate the concentration in μg/g, we used an equation obtained from a calibration curve that was performed previously with glucose concentrations going from 2.5 μg/mL to 40 μg/mL.

### 2.9. Enzyme Activities Related to Oxidative Stress

Enzyme activity was determined on a buffered enzyme extract (10 mg of leaf powder in 1 mL of potassium phosphate buffer 0.1 M pH 7 supplemented with 2 mM Phenylmethyl sulfonyl fluoride), manipulated at 4 °C. After sonication (10 min) and 10 min centrifugation at 14,000 rpm, soluble proteins were determined in the supernatant. Then, total protein was determined by mixing 50 μL of supernatant with 250 μL of Bradford reagent, using ELISA 96-well plates. After incubating at room temperature for 30 min, absorbance at 595 nm was determined in a plate reader (Heales. MB-580). A calibration curve was constructed from commercial BSA (0.05 and 2 mg/mL) to interpolate absorbances and determine protein concentration.

A spectrophotometer was used to determine enzyme activities related to scavenging of ROS in the supernatant, namely, ascorbate peroxidase (APX, EC 1.11.1.11), guaiacol peroxidase (GPX, EC 1.11.1.7), and glutathione reductase (GR, EC 1.6.4.2), all expressed as μmol/mg protein and min.

APX was measured as in Garcia-Limones et al. [26]. First, the enzyme extract (100 μL) was mixed with 860 μL of potassium phosphate buffer (50 mM pH 7.0) and 120 μL of 2.5 mM sodium ascorbate. The reaction was started by adding 120 μL of 50 mm H_2_O_2_. A decrease in absorbance at 290 nm was indicative of the oxidation of ascorbate. Activity was calculated using the extinction coefficient of 2.8 mM^−1^/cm.

GPX was measured as in Garcia-Limones et al. [26]. A mixture of enzyme extract (200 μL), 880 μL of 100 mM potassium phosphate buffer (pH 6.5), and 120 μL of 150 mM guaiacol was prepared. Reaction was started by adding 1 μL of H_2_O_2_ and the increase in absorbance at 470 nm indicated oxidation of guaiacol. Activity was calculated using the extinction coefficient of 26.6 mM^−1^/cm.

GR was measured as in Garcia-Limones et al. [26]. A mixture of enzyme extract (180 μL), 740 μL of potassium phosphate buffer (50 mM pH 7.5), 120 μL of 10 mM DNTB, 120 μL of 1 mM NADPH, and 120 μL of 10 mM oxidized glutathione in a final volume of 1.2 mL was prepared. Reaction was started by adding 100 μL of enzyme extract; oxidation of NADH was determined by the increase in A340. An extinction coefficient of 6.2 mM^−1^/cm was used to calculate activity. Blanks were prepared for each reaction by replacing the enzyme extract with buffer. In the case of GR activity, an additional blank without oxidized glutathione was also used to detect presence of other enzyme activities able to oxidize NADPH.

### 2.10. H_2_O_2_ Determination

The quantification of H_2_O_2_ was performed following the method of Shukla et al. [27]. Samples (0.2 g) of were suspended, and vortexed with 2.0 mL of 0.1% (*w*/*v*) trichloroacetic acid (TCA) in an ice bath. The homogenate was centrifuged at 10,000× *g* for 20 min. The supernatant (0.5 mL) was mixed with 0.5 mL of 10 mM of potassium phosphate buffer (pH 7.0) and 1 mL of potassium iodide. After 5 min, absorbance was measured spectrophotometrically at 390 nm. The amount of H_2_O_2_ formed was estimated from a standard curve (between 2 and 20 nmol) and expressed as nmol H_2_O_2_/g fresh weight (FW).

### 2.11. Malondialdehyde (MDA) Determination

Malondialdehyde (MDA) content was determined as described by Hu et al. [28]. Briefly, 225 mg of powder were added to 2 mL of trichloroacetic acid (TCA) 20% (*v*/*v*) and thiobarbituric acid (TBA) 0.5% (*v*/*v*). The reaction was incubated at 95 °C for 30 min and cooling to room temperature. Then, it was centrifuged at 6030× *g* for 20 min and the absorbance at 532 and 600 nm of the supernatant was measured spectrophotometrically. The concentration of MDA was determined as: MDA (nmol/g FW) = [(Abs 532 − Abs 600)]/(ε × FW), where FW is the plant fresh weight and ε the molar extinction coefficient (155 mM^−1^/cm). Results were expressed as μmol/g FW.

### 2.12. Total Phenolic Compounds

A methanolic extract was prepared with leaf powder (Section 2.7) and 80% methanol (1:9 *w*/*v*). After sonication for 10 min and centrifugation (4 °C, 5 min, 3.500 rpm), total phenolic compounds and flavanols were determined on the supernatant. Quantitative analysis of total phenolic compounds (TPC) was performed with Folin–Ciocalteau reagent (Sigma Aldrich, St Louis, MO, USA) [29] with modifications, using gallic acid as a standard (Sigma-Aldrich, St. Louis, MO, USA). A mixture of the supernatant (1 mL), 250 μL of a 2 N Folin–Ciocalteau reagent (Sigma-Aldrich), and 750 μL of 20% Na_2_CO_3_ solution was prepared. After incubating at room temperature for 30 min, a UV-Visible spectrophotometer (Biomate 5) was used to measure absorbance at 760 nm. A calibration curve was constructed with gallic acid. Results were expressed as mg gallic acid equivalent (GAE) per mg of fresh weight (FW).

### 2.13. RNA Extraction and RT-qPCR Analysis

Prior to RNA extraction, samples were ground to a fine powder with liquid nitrogen. Total RNA was isolated from each replicate with PureLink RNA Micro Kit (Invitrogen, Waltham, MA, USA), DNAase treatment included. RNA purity was confirmed using NanodropTM. A retrotranscription followed by RT-qPCR was performed.

The retrotranscription was performed using iScript tm cDNA Synthesis Kit (Bio-Rad, Hercules, CA, USA). All retrotranscriptions were carried out using a GeneAmp PCR System 2700 (Applied Biosystems, Foster City, CA, USA): 5 min 25 °C, 30 min 42 °C, 5 min 85 °C, and hold at 4 °C. Amplification was carried out with a MiniOpticon Real Time PCR System (Bio-Rad): 3 min at 95 °C and then 39 cycles consisting of 15 s at 95 °C, 30 s at 55 °C, and 30 s at 72 °C, followed by melting curve to check results. To describe the expression obtained in the analysis, cycle threshold (Ct) was used. Standard curves were calculated for each gene, and the efficiency values ranged between 90 and 110%. Results for gene expression were expressed as differential expression by the 2^−∆∆Ct^ method [30]. Actin gene was used as reference (housekeeping) gen. *P5CS* (Pyrroline-5-carboxylate synthase, related to proline biosynthesis), *NCED1* (9-cis-epoxycarotenoid dioxygenase, related to ABA biosynthesis), and *PM ATPase 1* (plasma membrane ATPase 1) genes were studied. Gene primers used are shown in Table 1.

### 2.14. Statistical Analysis

Student’s *t*-test was performed to check the statistical differences between control and PEG treatments. One-way ANOVA with replicates was used to check the statistical differences among treatments with PEG (PEG, PEG + N 5.12 and PEG + N 21.24) in all data obtained. Prior to ANOVA analysis, homoscedasticity and normality of the variance were checked with Statgraphics Plus 5.1 for Windows, meeting requirements for analysis. When significant differences appeared (*p* < 0.05), a Fisher test was used [31]. Principal components analysis (PCA) were performed with all parameters measured in root and aerial part separately. These analyses were performed using the software PAleontological STatistics (Past4).

## 3. Results

Net photosynthesis (Pn), transpiration (E), and water use efficiency are shown in Figure 2. The efficiency of water stress induced by PEG is evidenced by a significant decrease in net carbon fixation (Figure 2a) and transpiration (Figure 2b) as compared to naïve controls; however, it did not affect WUE (Figure 2c). However, bacterial-inoculated plants, challenged with PEG, significantly increased net C fixation compared to PEG controls (Figure 2a), reaching similar activity to naïve controls. The effects on transpiration were different in each bacterial strain; while N5.12 showed similar values to PEG controls, N21.24 significantly increase transpiration up to the values of naïve controls (Figure 2b). Consistently, WUE (Pn/E) increased significantly only with N5.12 (Figure 2c).

Photosynthetic efficiency measured through the chlorophyll fluorescence emitted by photosystem II is shown in Figure 3. PEG treatment induced a significant increase in NPQ regarding control plants. The other parameters were not modified in PEG treatment with regard to control plants. In inoculated plants (PEG + N 5.12 and PEG + N 21.24), the maximum photosynthetic efficiency of photosystem II (Fv/Fm; Figure 3b) significantly increased with regard to PEG treatment. Effective quantum yield (ϕPSII; Figure 3c) was not modified by any treatment.

Photosynthetic pigments are shown in Figure 4. PEG treatment significantly decreased all pigments with regard to control. Plants inoculated with N 5.12 increased all pigments measured (chlorophyll a, chlorophyll b, and carotenoids) with regard to the other treatments. However, plants inoculated with N 21.24 showed the same concentrations as in PEG controls.

Compatible solutes were measured in roots and leaves separately (Figure 5). In all cases, PEG treatment significantly modified values with regard to control with different effects in each organ. In roots, glycine betaine (Figure 5a) and proline (Figure 5b) significantly increased while soluble sugars (Figure 5c) significantly decreased with regard to healthy controls. Bacteria significantly decreased glycine betaine and proline, while soluble sugar concentration was not affected compared to PEG plants in roots. In leaves, glycine betaine (Figure 5d) and soluble sugars (Figure 5f) increased in PEG plants compared to healthy controls, while proline (Figure 5e) decreased, all changes being significant. Bacterial inoculation caused an increase in glycine betaine while soluble sugars decreased, and proline responded differently to each strain, increasing only with N5.12; all modifications were significant.

Enzymatic activities related to oxidative stress are shown in Figure 6. In all cases, PEG treatment significantly modified values with regard to control, decreasing in roots, and increasing in leaves. In roots (Figure 6a–c), plants inoculated with N 21.24 provoked a significant decrease of guaiacol peroxidase and ascorbate peroxidase (Figure 6a,b), while N 5.12 only caused significantly changes in ascorbate peroxidase with regard to PEG treatment. In leaves (Figure 6d–f), bacterial inoculation significantly decreased all enzyme activities with regard to PEG treatment.

Oxidative stress markers (H_2_O_2_ and MDA) are shown in Figure 7; a different behavior was detected in each organ. In roots, PEG significantly reduced H_2_O_2_ concentration (Figure 7a), while in leaves, a significant increase was detected regarding control (Figure 7c). Bacterial inoculation kept H_2_O_2_ levels the same as in PEG plants in roots (Figure 7a), while decreasing H_2_O_2_ concentration in leaves, reaching healthy control values (Figure 7c).

MDA followed a similar pattern as H_2_O_2_, significantly decreasing in roots and increasing in leaves due to PEG treatment (Figure 7b,d). Bacterial inoculation significantly reduced MDA levels in leaves (Figure 7d), while the response in roots was bacterial-dependent, with N5.12 increasing levels and N21.24 not affecting it, compared to PEG.

Total phenolic compounds in leaves are shown in Figure 8. PEG significantly increased phenolic compounds with regard to control, and bacterial inoculation caused a significant increase compared to PEG plants, reaching even higher values.

Figure 9 shows the principal component analysis (PCA) performed with the results of the parameters measured in the root (Figure 9a) and with the parameters measured in the aerial part (Figure 9b). In roots, the samples separate into three groups: healthy controls separate from PEG treatments along axis I, that absorbs 76.6% of the variance, and PEG treatments separate along axis II, that accounts for 16.15% of the variance, grouping inoculated samples towards the negative values and non-inoculated ones towards the positive values. The loading factors responsible for this separation are glycine betaine towards the negative end of axis I, and enzyme activities towards the positive ends, while proline and MDA are relevant in the separation along axis II. In leaves, (Figure 9b), four groups appear, three along axis I, that account for 56.55% of the variance, and one along axis II, that accounts for 23.09% of the variance. Each group corresponds to an experimental treatment. PEG control separates towards the positive end of axis I, PEG + N5.12 towards the negative end of Axis I, and PEG + N21.24 towards the negative end of axis II, leaving controls in the middle. The loading factors that determine this separation are enzyme activities towards the positive end of axis I; chlorophyll, soluble sugars, and net photosynthesis towards the negative end (N5.12); and NPQ, transpiration, and MDA towards the negative end of axis II (N21.24).

Figure 10 shows the differential expression of the *P5CS*, *NCDE1*, and *PM ATPase* genes. The expression of proline and ABA-related genes in PEG plants was lower than in naïve controls, while the ATPase gene was overexpressed. While N5.12 plants showed significantly higher expression for all three genes, N21.24 plants only overexpressed ABA-related and ATPase genes.

## 4. Discussion

The results presented in this work confirm that the two bacterial strains assayed are able to improve plant adaptation to moderated water stress, targeting different aspects of plant physiological mechanisms involved in adaptation. Furthermore, each of the bacteria activates different mechanisms resulting in protection from water stress.

A critical point in drought experiments is the medium used and the way to achieve stress. Osmolovoskaya et al. [32] reviews the available experimental models, discussing their pros and cons, from which we have selected the soil-based method for this study. The obvious advantage of this model is the close similarity of experimental conditions to actual drought conditions in nature and agriculture. However, controlling the substrate water potential (Ψw) represents an essential limitation of this approach [32]. It is well-documented that PEG effectively decreases the medium Ψw, thereby disrupting the absorption of water by plant roots [33]. In terms of this approach, PEG concentrations between 5–20% (*w*/*v*) [34] or even 40% (*w*/*v*) [35] in the growth medium enables a stable decrease of Ψw during any desired period of time [36]. Therefore, setting the desired stress with PEG requires attention and the 10% PEG concentration was determined in a previous experiment from a range of PEG concentrations (5, 10, 15, 20, and 25%), since 15% PEG induced marked wilting symptoms. The results obtained show that a 10% PEG concentration is sufficient for plants to detect stress and respond to it, without phenotypical symptoms, achieving a moderate drought stress situation.

One of the primary plant responses to dehydration is stomata closure, which is intended to prevent transpiration-related water loss and is essential for the success of the drought avoidance strategy [37,38]. Since stomata closure disrupts the supply of parenchyma cells with carbon dioxide, drought ultimately negatively affects the efficiency of photosynthesis by inhibiting carbon assimilation and light reactions [6] as evidenced in PEG-treated plants. Bacterial inoculation increased carbon fixation compared to PEG-treated plants, reaching levels detected in healthy controls, although transpiration was affected differently; while N 21.24 increased transpiration, N 5.12 maintained similar values as PEG plants, resulting in a higher WUE. Plants treated with N 21.24 keep their stomata open but are not able to maintain their photosynthetic capacity as high as plants treated with N 5.12. This may be due to the clear degradation of pigments that has occurred both in the PEG treatment and in the plants inoculated with N 21.24. In contrast, in plants treated with N 5.12, chlorophyll degradation is not observed, and there is an increase in all pigments. Some works have shown that a high pigment content in drought stress situations makes plants more resistant to stress [39,40]. A decreased chlorophyll level is considered a symptom of oxidative stress and may be the result of pigment photo-oxidation and chlorophyll degradation [41], which has been related to an increase in ethylene under stress. The favorable effect on pigments registered with strain N 5.12 may be attributed to its ACC deaminase activity, as degradation of the precursor to obtain nitrogen will result in reducing the concentration of ethylene in the plant [42].

These effects found on photosynthesis and chlorophyll concentration have been described by other authors working with other bacterial genera and plants, which suggests that it may be a common mechanism of action in plants subjected to water stress and elicited with PGPRs. [43,44]. Regarding the chlorophyll content, Puangbut et al. [45] propose it as a good indicator to identify varieties resistant to water stress.

The fluorescence emission data from photosystem II indicate a significant increase in Fv/Fm in the inoculated plants and an increase in energy dissipation (NPQ) in the presence of PEG regardless of the presence or absence of the bacterium. On the other hand, plants inoculated with N 5.12 have significantly higher carotenoid values (carotenes + xanthophylls) than the other treatments. The relationship between the different types of xanthophylls and the NPQ is well-known [46]. Given that the NPQ does not increase significantly in this treatment (PEG + N 5.12) with respect to the other treatments, it can be deduced that the increase in carotenoids has probably not been due to an increase in xanthophylls (responsible for energy dissipation) but rather due to an increase in carotenoids in antennae. This has an impact on the improvement in the efficiency of the photosystems and in their capacity to capture energy, which is reflected in the increase in Fv/Fm. Plants inoculated with N 21.24 also improve the Fv/Fm parameter but must modify other mechanisms since they do not increase the concentration of photosynthetic pigments. This is a very important aspect in the use of PGPR, since each one can improve tolerance to stress in different ways, opening up a very wide range of possibilities [13].

Compatible osmolytes have a very important role in water stress situations. Of them, proline, glycine betaine, and soluble sugars are among the most important [47]. It is well-known that the role of glycine betaine and proline in plants exposed to saline soil is to protect plant cells from salt stress by osmotic adjustment, protein stabilization (RuBisCo), photosynthetic apparatus protection, the protection of chlorophylls, and the reduction of oxygen radical scavengers [48,49,50,51,52]. The capability of tomato to produce and accumulate glycine betaine is controversial [52,53,54], but our results demonstrate the production of glycine betaine both in the root and in the leaves, especially in leaves of tomato plants inoculated with bacteria. In the root, the concentration of glycine betaine is 10 times higher than in the leaves, and its function must be clearly different, behaving as an osmotic adjusting in the root and as a protector of the photosynthetic apparatus in the leaf, since the synthesis occurs in the chloroplast [8]. Proline concentration is significantly higher in the leaves of plants inoculated with N 5.12, contributing to the greater protection of the photosynthetic apparatus that these plants seem to have. Yasmin et al. [55] demonstrated how the use of a strain of *Pseudomonas* inoculated in *Arabidopsis* seedlings subjected to water stress improved their resistance by increasing the concentration of proline, glycine betaine, and chlorophyll concentration. The concentration of proline in the leaves of the plants inoculated with N5.12 is related to a significantly higher expression of the *P5CS* gene in these plants (Figure 10). This relationship is well described in the bibliography [56,57] and that is why this gene is used as a marker of the proline biosynthesis pathway. Although in some works, there are discrepancies about the relationship between the concentration of proline and the resistance of plants against water stress [58], in general, there is considerable agreement about its influence on the improvement of plant resistance under these conditions; it can be used as a marker in the search for resistant varieties [59,60]. Proline not only has an important role in increasing the osmotic potential [60], but also is considered a powerful antioxidant that scavenges ROS that are produced in situations of water stress.

The production of reactive oxygen species (ROS) under stress conditions is one of the first consequences derived from such situations. Its elimination is vital to avoid oxidative stress [61]. PEG-treated plants responded to water stress even with higher enzymatic activities than the control plants, but they have not succeeded in reducing the concentration of H_2_O_2_ or MDA, while inoculated plants significantly decreased both oxidative stress markers (MDA and H_2_O_2_) in the leaf (Figure 7c,d). The improvement of oxidative stress in plants inoculated with PGPR is something well-known and demonstrated; many of these strains are capable of ameliorating oxidative stress either by preventing ROS formation or by triggering ROS scavenging mechanisms in different ways, either targeting antioxidant enzymes and metabolites. In this case, it does not seem that enzymatic activities are responsible for this (Figure 6), so other mechanisms may be contributing to it, such as the proline concentration (Figure 5f) which, in addition to being an osmoprotector, has a powerful antioxidant effect [62], or others, such as the production of phenolic compounds (Figure 8); in any case, the protective effect exerted by the strains on the inoculated plants is beyond any doubt.

ABA is a hormone that plays a major role in stomatal closure in situations of water stress and, in general, in different abiotic stress situation. The PEG-treated plants decreased carbon fixation and transpiration significantly with respect to the control, indicating a clear effect in stomatal closure to prevent water loss. *NCDE1* gene expression (Figure 10) in PEG control plants was lower than in bacterial-inoculated plants, consistent with their naïve state facing the stress opposite to the primed-post challenge status on bacterial-treated plants [11]. Bacterial-treated plants showed different behavior in transpiration, while N 5.12 maintained similar transpiration as PEG, and N 21.24 increased transpiration, meaning that the stomata remained open under stress. This is consistent with the expression of the *NCDE1* gene, involved in ABA synthesis. Plants treated with N 5.12 showed higher expression, supporting ABA-mediated stomata closure. However, N 21.24 overexpresses NCDE1, suggesting that ABA-mediated stomata closure is counteracted by some other mechanism. The activation of ABA-mediated signaling for stomatal closure occurs through the binding of ABA to its PYR/PYL/RCAR receptors [9,63]. It is possible that N 21.24 would be somehow blocking this union. On the other hand, strain N 21.24 causes a greater differential expression of *PM ATPase 1* (plasma membrane ATPase 1), which could reinforce the maintenance of the open stomata, since these pumps play an important role in maintaining the turgor of the guard cells [64], as well as being related to the transport of ions through the membranes [65].

As evidenced in the PCA (Figure 9), PEG control plants have detected the stress and have responded to it, and the inoculated plants have done so in a different way. However, the effect of the PGPR has been very different at the root level than at the foliar level. The systemic effect has been more noticeable, and a clear difference is also seen between the changes produced in the plant by the two bacteria.

## 5. Conclusions

The scarcity of water in agricultural systems is going to become one of the most important problems in the coming years in food production. Research in this field is shown to be key if we want to produce food for a growing population. In this work, we have addressed the issue through the use of PGPRs as a biotechnological tool capable of eliciting the innate systems of plants to resist water stress conditions without losing productive capacity. The bacteria used in this work have shown that their use can be very effective. Both have been able to improve photosynthetic efficiency under moderate water stress conditions, one of them (N 5.12) increasing the concentration of photosynthetic pigments, compatible solutes, and phenols, which led to less oxidative stress (measured by the MDA marker). Both strains modify the physiology of plants in different ways, making them adapt to stress conditions. The use of this type of bacteria could become a viable and economical alternative in agricultural systems exposed to water stress, reducing production losses that occur in these situations. Experiments are currently being carried out under real growing conditions to determine if the observed effects hold up under real crop conditions.

## Figures and Tables

**Figure 1 biology-12-00901-f001:**
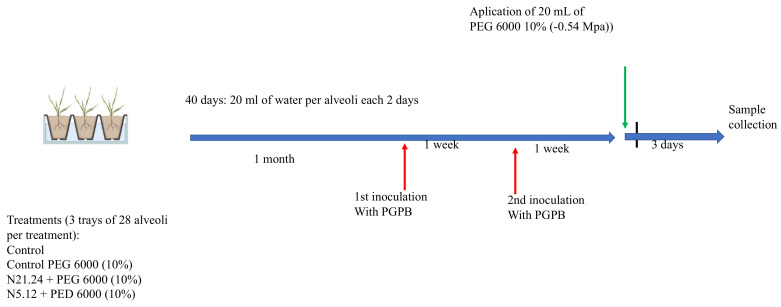
Experimental design.

**Figure 2 biology-12-00901-f002:**
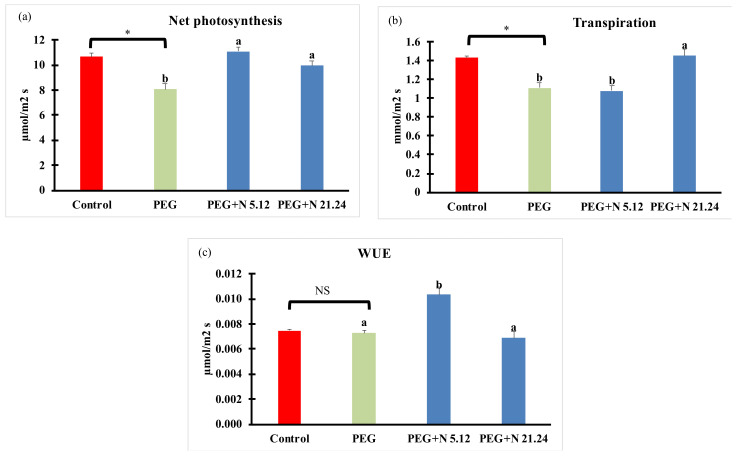
(**a**) Net photosynthesis (Pn); (**b**) transpiration (E); and (**c**) water used efficiency (WUE). Values are means +/− standard error (SE) of 15 biological replicates, each corresponding to different plants. Asterisk (*; significant differences) or NS (non-significant differences) indicate Student’s *t*-test results between Control and PEG treatments (*p* < 0.05). Different letters indicate significant differences in ANOVA test among PEG, PEG + N 5.12, and PEG + N 21.24 (*p* < 0.05).

**Figure 3 biology-12-00901-f003:**
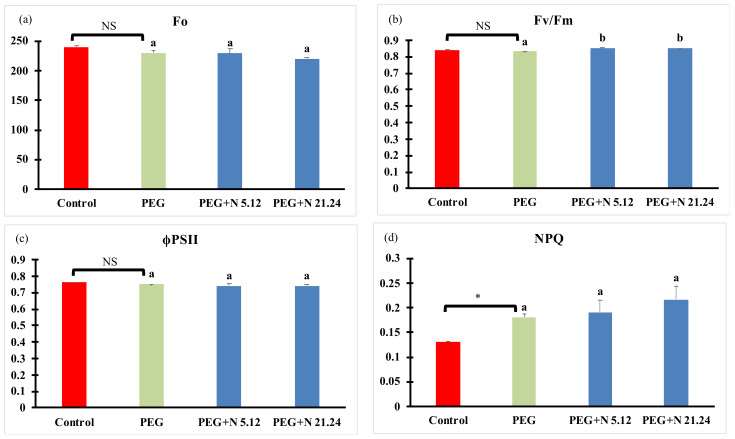
(**a**) Minimal fluorescence (Fo); (**b**) maximum photosynthetic efficiency (Fv/Fm); (**c**) effective PSII quantum yield (ϕPSII); and (**d**) non-photochemical quenching (NPQ). Values are means +/− standard error (SE) of 15 biological replicates, each corresponding to different plants. Asterisk (*; significant differences) or NS (non-significant differences) indicate Student’s *t*-test results between Control and PEG treatments (*p* < 0.05). Different letters indicate significant differences in ANOVA test among PEG, PEG + N 5.12, and PEG + N 21.24 (*p* < 0.05).

**Figure 4 biology-12-00901-f004:**
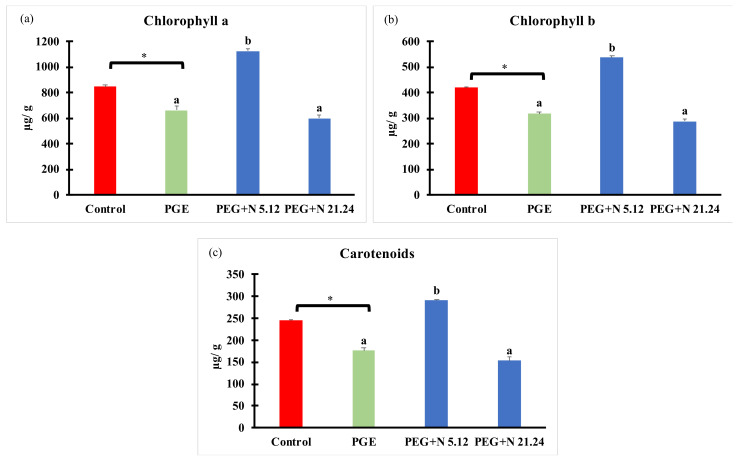
Photosynthetic pigments concentration: (**a**) chlorophyll a; (**b**) chlorophyll b; and (**c**) carotenoids. Values are means +/− standard error (SE) of 3 biological replicates, each replicate consisted of 10 plants. Asterisk (*; significant differences) indicate Student’s *t*-test results between Control and PEG treatments (*p* < 0.05). Different letters indicate significant differences in ANOVA test among PEG, PEG + N 5.12, and PEG + N 21.24 (*p* < 0.05).

**Figure 5 biology-12-00901-f005:**
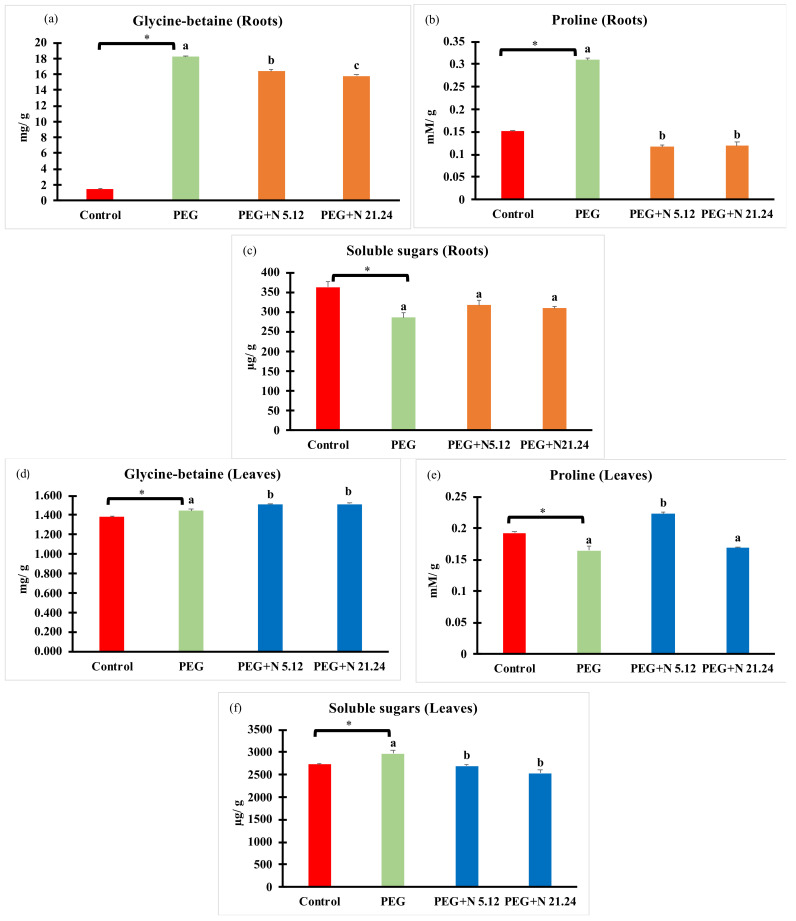
Compatible solutes: glycine betaine, proline, and soluble sugars measured in roots (**a**,**b**,**c**), and in leaves (**d**,**e**,**f**). Values are means +/− standard error (SE) of 3 biological replicates, each replicate being a pool of 10 plants. Asterisk (*; significant differences) indicate Student’s *t*-test results between Control and PEG treatments (*p* < 0.05). Different letters indicate significant differences in ANOVA test among PEG, PEG + N 5.12, and PEG + N 21.24 (*p* < 0.05).

**Figure 6 biology-12-00901-f006:**
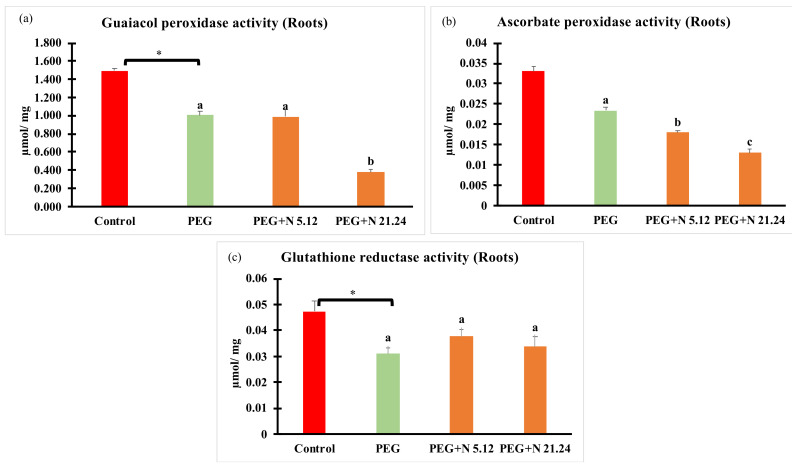
Enzyme activities related to oxidative stress: (**a**) guaiacol peroxidase activity in root; (**b**) guaiacol peroxidase activity in leaves; (**c**) ascorbate peroxidase activity in root; (**d**) ascorbate peroxidase activity in leaves; (**e**) glutathione reductase activity in root; and (**f**) glutathione reductase activity in leaves. Values are means +/− standard error (SE) of 3 biological replicates, each replicate being a pool of 10 plants. Asterisk (*; significant differences) indicate Student’s *t*-test results between Control and PEG treatments (*p* < 0.05). Different letters indicate significant differences in ANOVA test among PEG, PEG + N 5.12, and PEG + N 21.24 (*p* < 0.05).

**Figure 7 biology-12-00901-f007:**
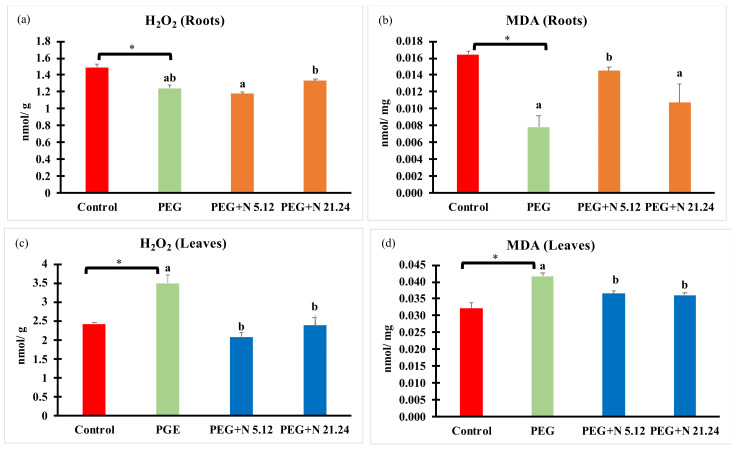
H_2_O_2_ and malondialdehyde concentration in root (**a**,**b**) and in aerial part (**c**,**d**). Values are means +/− standard error (SE) of 3 biological replicates, each replicate being a pool of 10 plants. Asterisk (*; significant differences) indicate Student’s *t*-test results between Control and PEG treatments (*p* < 0.05). Different letters indicate significant differences in ANOVA test among PEG, PEG + N 5.12, and PEG + N 21.24 (*p* < 0.05).

**Figure 8 biology-12-00901-f008:**
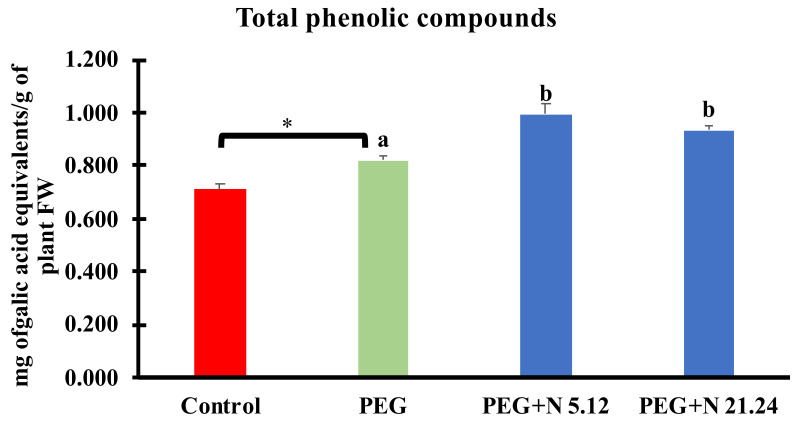
Total phenolic compounds in aerial part. Values are means +/− standard error (SE) of 3 biological replicates, each replicate being a pool of 10 plants. Asterisk (*; significant differences) indicate Student’s *t*-test results between Control and PEG treatments (*p* < 0.05). Different letters indicate significant differences in ANOVA test among PEG, PEG + N 5.12, and PEG + N 21.24 (*p* < 0.05).

**Figure 9 biology-12-00901-f009:**
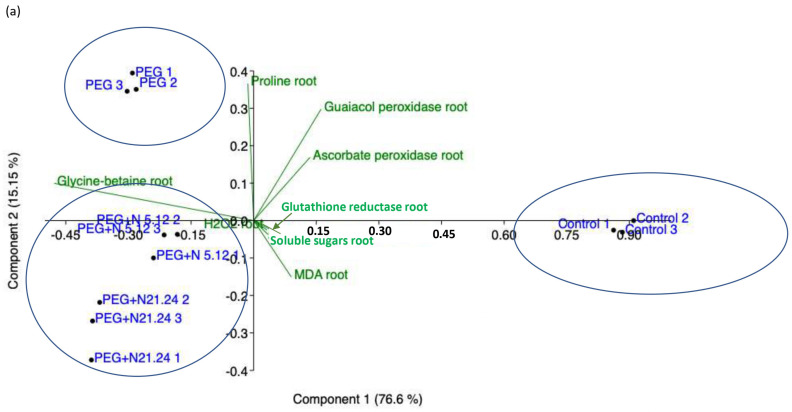
Principal component analysis (PCA) made with data from root material (**a**), and aerial part material (**b**).

**Figure 10 biology-12-00901-f010:**
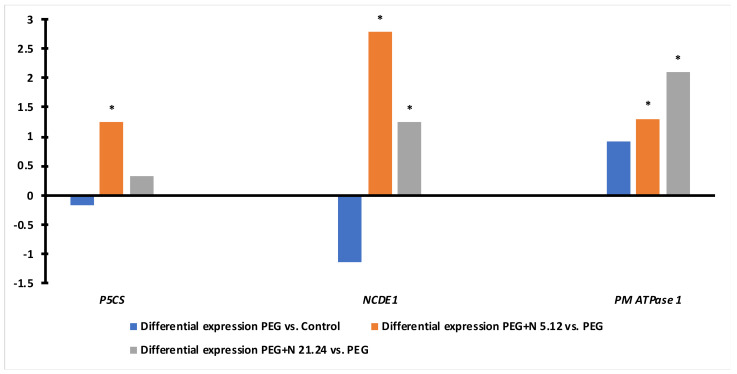
Differential expression of genes *P5CS* (Pyrroline-5-carboxylate synthase, related to proline biosynthesis), *NCED1* (9-cis-epoxycarotenoid dioxygenase, related to ABA biosynthesis), and *PM ATPase 1* (plasma membrane ATPase 1) in leaves comparing naïve PEG versus naïve controls (blue bars), and N5.12+PEG vs PEG (orange bars) and N21.24+PEG vs PEG (grey bars). Asterisks (*) indicate significant differences.

**Table 1 biology-12-00901-t001:** Forward and reverse primers used in qPCR analysis.

	Forward Primer	Reverse Primer
*Ls Actin*	5′GGAAAAGCTTGCCTATGTGG	5′CCTGCAGCTTCCATACCAAT
*Ls P5CS*	5′TGCTCAACAGGCCGGATATG	5′AAAGTGTGACCAAGGGGCTC
*Ls NCDE1*	5′CTGCTTCTTCCCAAGCTATC	5′ACCTGTTCCACCACAAGGAC
*Ls H+ ATPase plasma membrane*	5′CGAAGGATAGGGTCAAACCA	5′AGCCACCAAGAACAACTCCA

## Data Availability

There are no available datasets analyzed or generated during the study.

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
