# Peer review of "Physiological and Genetic Modifications Induced by Plant-Growth-Promoting Rhizobacteria (PGPR) in Tomato Plants under Moderate Water Stress"

_biology, 2023, doi:10.3390/biology12070901_

Round 1
Reviewer 1 Report
Comments to the Author
The manuscript on “Physiological and genetic modifications induced by two plant growth promoting rhizobacteria (PGPR) in tomato plants under moderate water stress”. Authors have done extensive research on two plant growth-promoting rhizobacteria (PGPR) Pseudomonas sp. (N 5.12 and N 21.24). The manuscript can be accepted for publication.
Interestingly, authors have tested stress-related physiological and biochemical parameters that can help tomato plants to sustain water stress. The authors should have evaluated tomato plant morphometric data treated with PGPR with regard to plant weight, yield, etc.
Fig. 5d typo error in PEG.
Line 392 typo error dependent
Line 124 typo error ..
In the title remove the word "two."
Author Response
First of all, we would like to thank you for your constructive criticisms, which have substantially improved the quality of the manuscript. We have reviewed and modified all changes suggested by the reviewer.
Reviewer 2 Report
The present manuscript, "Physiological and genetic modifications induced by two plant growth promoting rhizobacteria (PGPR) in tomato plants under moderate water stress" is a study on the effect of PGPR on the physiological, metabolite, and genetic changes in tomato plants. The paper has an exciting theme and is an original and new contribution to the field. However, some questions need to be addressed for more clarification.
1. Abstract: The abstract is poorly structured and lacks quantitative/numerical data. Please include more significant results, a concluding remark, and the future scope of this study. Apart from this, many technical issues detected here, i.e., botanical names are not in ITALIC form, abbreviations in the full form not given, what are N5.12 and N21.24 not mentioned, etc. (Please see attached file).
2. Introduction: The language and style of the Introduction section must be improved. Many sentences are unclear, hard to understand, and confusing. Many technical errors are similar to the abstract, and many symbols and font types (not looking English font) are detected. Writing 2-3 paragraphs for an objective makes no sense and misleads the actual objective of the research. Therefore, please rewrite the scientific importance, research gaps, and objective concisely with more latest references.
3. Materials and methods: Many technical errors are similar to the abstract, and many symbols and font types (not looking like English font) are detected. Please check the attached file for more detailed comments.
5. Results: The results of this research should be written in a more interesting and logical way. Please clarify bacterial strains caused in this study are natural or mutated and their availability in soil.
Please clarify the photosynthesis and respiration in treatments PEG+ N21.24. As for the previous reports, transpiration is reduced in treatments than after photosynthesis, while your results are in contrast.
6. Discussion: The discussion should be more in-depth, and the studies should be compared in more detail.
7. The conclusion is unsatisfactory; please improve it, including the research highlight, the most significant results, a concluding remark on the study, and future investigations.
8. Most importantly, authors should pay more attention to typo/technical errors such as botanical names (for example, Pseudomonas sp., Arabidopsis thaliana, Solanum lycopersicum, etc. ) should be in "ITALIC." In between sentences, use of Capital letters in words (for example, line 111: Proline, 123: Seeds, and many more), proper spacing is not maintained (Line 176: spacing in the formula is incorrect), along with the use of symbols, proper fonts, and terms), proper units with (for example mol.m-2.s-1). Please carefully check similar errors throughout the paper and correct them according to standard units/forms/fonts/or types.
9. Please check all the question mark terms and sentences, names, terms, etc., in the entire paper.
Please refer to the review supporting file attached.

The language of the manuscript doesn't meet the minimum standards of publication. Many sentences are hard to understand, apart from some technical errors (Please see Attached review supporting file). Therefore, the author should pay more attention to the usage of grammar and punctuation. I suggest that the author send the manuscript to the English Language Editing Services or a native English speaker.
Author Response
First of all, we would like to thank you for your constructive criticisms, which have substantially improved the quality of the manuscript. The manuscript has been deeply revised by a native English speaker from the English department of our university. The English speaker has made a lot of changes.
The present manuscript, "Physiological and genetic modifications induced by two plant growth promoting rhizobacteria (PGPR) in tomato plants under moderate water stress" is a study on the effect of PGPR on the physiological, metabolite, and genetic changes in tomato plants. The paper has an exciting theme and is an original and new contribution to the field. However, some questions need to be addressed for more clarification.
- Abstract: The abstract is poorly structured and lacks quantitative/numerical data. Please include more significant results, a concluding remark, and the future scope of this study. Apart from this, many technical issues detected here, i.e., botanical names are not in ITALIC form, abbreviations in the full form not given, what are N5.12 and N21.24 not mentioned, etc. (Please see attached file).
The abstract has been reviewed and modified following the reviewer's suggestions.
- Introduction: The language and style of the Introduction section must be improved. Many sentences are unclear, hard to understand, and confusing. Many technical errors are similar to the abstract, and many symbols and font types (not looking English font) are detected. Writing 2-3 paragraphs for an objective makes no sense and misleads the actual objective of the research. Therefore, please rewrite the scientific importance, research gaps, and objective concisely with more latest references.
The introduction has been reviewed by a native speaker, all typos have been corrected and the last paragraph has been clarified so that the objective is clear and some references that were a bit outdated have been changed (15-19).
Materials and methods: Many technical errors are similar to the abstract, and many symbols and font types (not looking like English font) are detected. Please check the attached file for more detailed comments.
We are sorry for all the typographical errors that were in this section, it happened when we passed the content to the template and we did not realize it.
- Results: The results of this research should be written in a more interesting and logical way. Please clarify bacterial strains caused in this study are natural or mutated and their availability in soil.
The way of describing the results seemed to us the best way to understand the work. First we describe and compare the control without stress against the control with hydric stress (PEG treatment) so that it is clear that the stress has been detected or not by the plant, to then compare the treatments that have been exposed to stress without bacteria or with bacteria and to be able to make clear the effect of the PGPR and the priming effect. Even so, we have made some changes in the results section to clarify some of them.
We have included the data on the strains in material and methods in the strain used section.
Please clarify the photosynthesis and respiration in treatments PEG+ N21.24. As for the previous reports, transpiration is reduced in treatments than after photosynthesis, while your results are in contrast.
We have modified the sentence to clarify it.
- Discussion: The discussion should be more in-depth, and the studies should be compared in more detail.
We have included more papers to be able to compare and discuss the results in more depth.
- The conclusion is unsatisfactory; please improve it, including the research highlight, the most significant results, a concluding remark on the study, and future investigations.
We have rewritten the conclusion section.
- Most importantly, authors should pay more attention to typo/technical errors such as botanical names (for example, Pseudomonas sp., Arabidopsis thaliana, Solanum lycopersicum, etc. ) should be in "ITALIC." In between sentences, use of Capital letters in words (for example, line 111: Proline, 123: Seeds, and many more), proper spacing is not maintained (Line 176: spacing in the formula is incorrect), along with the use of symbols, proper fonts, and terms), proper units with (for example mol.m-2.s-1). Please carefully check similar errors throughout the paper and correct them according to standard units/forms/fonts/or types.
We have done a thorough review and we think all typo/technical errors have been fixed.
- Please check all the question mark terms and sentences, names, terms, etc., in the entire paper.
Please refer to the review supporting file attached.
Comments on the Quality of English Language
The language of the manuscript doesn't meet the minimum standards of publication. Many sentences are hard to understand, apart from some technical errors (Please see Attached review supporting file). Therefore, the author should pay more attention to the usage of grammar and punctuation. I suggest that the author send the manuscript to the English Language Editing Services or a native English speaker.
The manuscript has been deeply revised by a native English speaker from the English department of our university. The English speaker has made a lot of changes.
Reviewer 3 Report
Article "Physiological and genetic modifications induced by two plant growth promoting rhizobacteria (PGPR) in tomato plants under moderate water stress" study has been planned and performed in nice way. Data generates is also fine with respect to the study finding requires.
I have serious concern regarding typing error, manuscript has a large number of typing error like H2O2 writing style, similar mistakes are present in large number. Additionally English is poor which reduces the scientific soundness of the manuscript. Quality of figures are poor that must should be enhanced. In addition to this i have found that reference formatting is not in the style of journal guidelines. More references should be added in introduction and discussion section should be supported with more relevent references.
Article "Physiological and genetic modifications induced by two plant growth promoting rhizobacteria (PGPR) in tomato plants under moderate water stress" study has been planned and performed in nice way. Data generates is also fine with respect to the study finding requires.
I have serious concern regarding typing error, manuscript has a large number of typing error like H2O2 writing style, similar mistakes are present in large number. Additionally English is poor which reduces the scientific soundness of the manuscript. Quality of figures are poor that must should be enhanced. In addition to this i have found that reference formatting is not in the style of journal guidelines. More references should be added in introduction and discussion section should be supported with more relevent references.
Author Response
First of all, we would like to thank you for your constructive criticisms, which have substantially improved the quality of the manuscript. The manuscript has been deeply revised by a native English speaker from the English department of our university. The English speaker has made a lot of changes.
Article "Physiological and genetic modifications induced by two plant growth promoting rhizobacteria (PGPR) in tomato plants under moderate water stress" study has been planned and performed in nice way. Data generates is also fine with respect to the study finding requires.
I have serious concern regarding typing error, manuscript has a large number of typing error like H2O2 writing style, similar mistakes are present in large number. Additionally English is poor which reduces the scientific soundness of the manuscript. Quality of figures are poor that must should be enhanced. In addition to this i have found that reference formatting is not in the style of journal guidelines. More references should be added in introduction and discussion section should be supported with more relevent references.
We are very sorry for all the typographical errors that exist, when passing the manuscript to the template those errors occurred and we did not realize it. It has been revised and we believe that they are all corrected.
Reviewer 4 Report
Though you have got many results about two plant growth promoting rhizobacteria (PGPR) in tomato plants under moderate water stress.
But there are many mistakes in writing and some missing according to the journal.
Like,
Where is the “Simple Summary”?
The references did not follow the requirements of the journal.
I just give some examples of errors.
What is PEG? Seems not mention this.
The sentence of L10-L12 in the Abstract not clear enough.
It is strange that in L35 there is just one sentence as a paragraph.
L48 H2O2 should be H2O2., L182 H2SO4 should be H2SO4. And many other thorough the article.
In Figure 2 why not compare this 4?
L367, 382, 405 what is “student¬¥s t-test”?
Figure 8. where is the Vertical axis?
The sentence of L10-L12 in the Abstract not clear enough.
Author Response
First of all, we would like to thank you for your constructive criticisms, which have substantially improved the quality of the manuscript. The manuscript has been deeply revised by a native English speaker from the English department of our university. The English speaker has made a lot of changes.
Though you have got many results about two plant growth promoting rhizobacteria (PGPR) in tomato plants under moderate water stress.
But there are many mistakes in writing and some missing according to the journal.
Like,
Where is the “Simple Summary”?
We have included the “simple summary” in the revised version
The references did not follow the requirements of the journal.
References have been ordered and cited in the journal format.
I just give some examples of errors.
What is PEG? Seems not mention this.
It has been indicated in the abstract that it was where it appeared for the first time.
The sentence of L10-L12 in the Abstract not clear enough.
It is strange that in L35 there is just one sentence as a paragraph.
L48 H2O2 should be H2O2., L182 H2SO4 should be H2SO4. And many other thorough the article.
In Figure 2 why not compare this 4?
For none of the measured parameters were the 4 treatments compared by ANOVA. Only the stress treatments were compared using this statistical technique. The control without stress was compared with the control with stress (PEG) using a student's t-test. By comparing the control with the stressed treatment using the student's t-test, we wanted to know if the stress was sufficient for the plants to have detected it. We could have done an ANOVA with the 4 treatments, but it is likely that in some cases we would have lost the statistical effect of the bacteria, as the true control of the bacteria treatments is the PEG, not the unstressed control.
L367, 382, 405 what is “student¬¥s t-test”?
Figure 8. where is the Vertical axis?
We are very sorry for all the typographical errors that exist, when passing the manuscript to the template those errors occurred and we did not realize it. It has been revised and we believe that they are all corrected
Comments on the Quality of English Language
The sentence of L10-L12 in the Abstract not clear enough
This sentence has been changed.
Round 2
Reviewer 2 Report
Manuscript can be accepted
Author Response
Dear reviewer, thank you very much for your comments, I am glad to know that the modifications made in the manuscript have been enough to consider that the work can be published
Reviewer 4 Report
It's much better than last version.
I think it's ok now.
Author Response

(The authors gave the same response as above.)
